# Development and Validation of the Predictive Model for the Differentiation between Vestibular Migraine and Meniere’s Disease

**DOI:** 10.3390/jcm11164745

**Published:** 2022-08-14

**Authors:** Dan Liu, Zhaoqi Guo, Jun Wang, E Tian, Jingyu Chen, Liuqing Zhou, Weijia Kong, Sulin Zhang

**Affiliations:** 1Department of Otorhinolaryngology, Union Hospital, Tongji Medical College, Huazhong University of Science and Technology, Wuhan 430022, China; 2Institute of Otorhinolaryngology, Union Hospital, Tongji Medical College, Huazhong University of Science and Technology, Wuhan 430022, China; 3Key Laboratory of Neurological Disorders of Education Ministry, Tongji Medical College, Huazhong University of Science and Technology, Wuhan 430022, China

**Keywords:** vestibular migraine, Meniere’s disease, predictive model, differential diagnosis, clinical features, auditory-vestibular function

## Abstract

(1) Background: Vestibular migraine (VM) and Meniere’s disease (MD) share multiple features in terms of clinical presentations and auditory-vestibular dysfunctions, e.g., vertigo, hearing loss, and headache. Therefore, differentiation between VM and MD is of great significance. (2) Methods: We retrospectively analyzed the medical records of 110 patients with VM and 110 patients with MD. We at first established a regression equation by using logistic regression analysis. Furthermore, sensitivity, specificity, accuracy, positive predicted value (PV), and negative PV of screened parameters were assessed and intuitively displayed by receiver operating characteristic curve (ROC curve). Then, two visualization tools, i.e., nomograph and applet, were established for convenience of clinicians. Furthermore, other patients with VM or MD were recruited to validate the power of the equation by ROC curve and the Gruppo Italiano per la Valutazione degli Interventi in Terapia Intensiva (GiViTI) calibration belt. (3) Results: The clinical manifestations and auditory-vestibular functions could help differentiate VM from MD, including attack frequency (X5), phonophobia (X13), electrocochleogram (ECochG) (X18), head-shaking test (HST) (X23), ocular vestibular evoked myogenic potential (o-VEMP) (X27), and horizontal gain of vestibular autorotation test (VAT) (X30). On the basis of statistically significant parameters screened by Chi-square test and multivariable double logistic regression analysis, we established a regression equation: P = 1/[1 + e^−(−2.269^
^× X5 − 2.395^
^× X13 + 2.141^
^× X18 + 3.949 × X23 + 2.798^
^× X27 − 4.275^
^× X30(1) − 5.811^
^× X30(2) + 0.873)^] (P, predictive value; e, natural logarithm). Nomographs and applets were used to visualize our result. After validation, the prediction model showed good discriminative power and calibrating power. (4) Conclusions: Our study suggested that a diagnostic algorithm based on available clinical features and an auditory-vestibular function regression equation is clinically effective and feasible as a differentiating tool and could improve the differential diagnosis between VM and MD.

## 1. Introduction

Meniere’s disease (MD) is a peripheral vestibular disorder that causes episodic vertigo and fluctuating hearing loss, tinnitus, and aural fullness. The prevalence of MD roughly stands somewhere between 34–190 per 100,000 [1]. The establishment of vestibular migraine (VM) diagnosis entails the presence of vertigo attacks associated with migrainous symptoms, such as headache and photophobia, among others [2]. VM represents one of the most common vestibular disorders and afflicts up to 1% of the general population [3] and 11% of patients visiting dizziness clinics [4]. Inevitably, some patients have episodes of vertigo that have the features of both diseases [5]. Though different mechanisms have been proposed, distinguishing between these two conditions remains challenging [6]. It has been well accepted that some patients might suffer from both diseases [7]. It has been reported that 30% of patients with vestibular migraine have no headache, and 51% of patients with Meniere’s disease experience migraine [8]. The vestibular symptoms of VM tend to be indistinguishable from those of MD. Moreover, it should be pointed out that the treatment approaches of the two conditions are different, and the non-targeted treatment without clear differential diagnosis might lead to even more unfavorable prognosis. Therefore, identifying measures or markers that differentiate VM from MD is an urgent task for both researchers and clinicians.

The current diagnostic criteria for MD and VM developed by the Classification Committee of the Bárány Society are mainly based on subjective complaints but lack objective measures [1,2]. To date, a few studies have employed some objective tests for the differential diagnosis of MD and VM. Multiple studies have found that loss of VOR, as detected by the caloric test, is more common and severe in MD than in VM; vHIT can provide additional information, and the two tests can be used in tandem. Vestibular evoked myogenic potentials (VEMPs) have been proposed as a marker for the differential diagnosis between MD and VM [9]. What is more, EH is a common pathology shared by both MD and VM, as revealed by gadolinium-contrasted MRI, and MD patients present EH much more frequently than their VM counterparts [10,11]. Two studies based on the same VM diagnostic criteria made an explicit comparison between MD and VM with regards to physical examination results [7,12]. Both studies found that abnormal headshake nystagmus (HSN) and abnormal vibration-induced nystagmus (VIN) are more frequent in the MD population. Murofushi et al. [5] suggested the term VM/MD overlapping syndrome to describe the cases where it was not possible to distinguish between MD and VM. Separate identification of VM and MD is the premise for diagnosing the presence of comorbidity. In addition, the application of artificial intelligence may lead to development of novel strategies for differential diagnosis of vertigo and dizziness [13,14]. Patients with MD and patients with VM are treated differently, and therefore, more accurate differential diagnosis of these two disorders should help to avoid mismanagement. Until now, no ascertained indicators are available that can be used for confirming or refuting the diagnosis of VM or MD.

Up to now, there exist no simple strategies for differentiating VM from MD. Misdiagnosis of VM or MD leads to erroneous management in the subsequent treatment of these two conditions. Hence, in this study, we used more comprehensive clinical features and auxiliary examination findings to establish a predictive model to facilitate the screening and the diagnosis of VM and MD for clinicians and medical care providers.

## 2. Materials and Methods

### 2.1. Data Collection and Processing

A total of 110 cases of VM and 110 cases of MD were retrospectively analyzed from the Department of Otorhinolaryngology, Union Hospital, Tongji Medical College, Huazhong University of Science and Technology, Wuhan, China, from February 2020 to February 2022.

Inclusion criteria: Diagnostic criteria of VM and probable VM were jointly formulated; the Committee for Classification of Vestibular Disorders of the Barany Society and the International Headache Society (IHS) developed a consensus document with diagnostic criteria that was included in the appendix of the new international classification of headache disorders (ICHD)-3 beta version [2]. The diagnostic criteria for MD cover two categories—definite MD and probable MD—and the criteria were jointly formulated by the Classification Committee of the Barany Society and some other scientific societies [1]. All patient records were reviewed by two specialists to decide if a patient met the diagnostic criteria of the definite or probable MD or VM for inclusion in the study.

Exclusion criteria: (1) having bilateral MD or family history and autoimmune disease history; (2) having reported overlapping symptoms of MD and VM or if satisfying the diagnostic criteria for both disorders based on their history; (3) having further vestibular or neurological disorders (e.g., benign paroxysmal positional vertigo, vestibular paroxysmia, inner and outer labyrinthine fistula, vestibular neuritis, vestibular schwannoma, cerebellar ataxia, extrapyramidal motor disorders, dementia, multiple sclerosis, stroke) or middle ear disease (e.g., cholesteatoma, otosclerosis, chronic otitis media, tympanic effusion); (4) having a history of ear surgery, brain surgery, or concussion; (5) having metabolic diseases or mental illnesses; (6) pregnant women and age < 18 years.

Then, we prospectively recruited 28 patients with MD and 28 patients with VM in the same center at different periods as validation groups, whose inclusion and exclusion criteria were mentioned above. Informed consent was obtained from all participants before their enrollment.

All the data pertaining to these patients were collected, the medical records of these patients were reviewed, and information regarding their demography, clinical manifestations, and auxiliary examinations was retrieved. The data were entered into EpiData 3.1 (The Epidata Association, Odense, Denmark) statistical software, which was used for database design and data entry [15]. Clinical variables included descriptions of vertigo/dizziness, illness duration, visual motion, nausea and vomiting, hearing loss and other otologic symptoms (e.g., tinnitus), headache or migraine, and personal and family neurotologic histories. Audiometric-vestibular assessments included pure tone audiometry (PTA), otoacoustic emission (OAE), electrocochleogram (ECochG), head-shaking test (HST), videonystagmography (VNG), caloric tests, vestibular evoked myogenic potentials (VEMPs), video-head impulse test (v-HIT), vestibular autorotation test (VAT), tests of sensory integration and balance, such as the sensory organization test (SOT). Somatic Self-rating Scale (SSS), Generalized Anxiety Disorder (GAD-7), and The Patient Health Questionnaire-9 (PHQ-9) were used for the assessment of patient self-reported dizziness-related handicap, anxiety, and depression, respectively. Patients were removed from the research if 2/3 of their examination items of the overall data were missing or unavailable [16,17]. Subsequently, the data were systematically imported into a Microsoft Excel chart. Each parameter or feature was assigned a value for subsequent logistic regression. The methodology and parameters of the tests are shown in Table 1.

This study was approved by the Ethics Committee of Union Hospital, Tongji Medical College, Huazhong University of Science and Technology, Wuhan, China (NO. 20210873). All procedures performed in the studies involving human participants were in strict accordance with the ethical standards of the institutional and/or national research committee and with the 1964 Helsinki Declaration and its later amendments or comparable ethical standards.

### 2.2. Statistical Analysis

Data were entered into EpiData 3.1 database and then analyzed using SPSS 23.0 (IBM, Armonk, NY, USA). Univariate Chi-square test (UCST) was applied to determine statistically significant variables that might be important in the differentiation of VM from MD. *p* < 0.05 was considered statistically significant. Then, statistically significant parameters were subjected to multivariable double logistic regression analysis and a regression equation (mathematical model) was established. Confidence intervals and ORs of the significant parameters were calculated.

Each significant parameter, separately or in combination with other parameters, was further analyzed for sensitivity, specificity, positive predictive value (PPV), negative predictive value (NPV), and accuracy in the diagnosis of VM or MD. Discriminative power of the regression model was verified by ROC curves. Then, visualization tools, i.e., nomographs and applets, were established through R software and Excel, respectively. ROC curve (C-index) and GiViTI calibration belt were used to estimate the discriminative power and calibrating ability of the prediction model in both the model construction group and validation group through R software (R, A Language and Environment for Statistical Computing, R Core Team, R Foundation for Statistical Computing, Vienna, Austria. 2021, URL http://www.R-project.org, R version 4.0.5, accessed on 9 May 2021).

## 3. Results

### 3.1. Demographics, Past History, and Auxiliary Examination Findings of VM and MD

A database involving a total of 220 patients was established and consisted of several parts (i.e., baseline data, clinical manifestations, auxiliary examination results). Demographics, clinical features, and auxiliary examination findings of VM and MD patients were recorded in all patients, and a value was assigned to each variable. Features in VM and MD were analyzed and screened to search for differentiating ones by using the univariate Chi-square test. We found that illness duration (*p* = 0.005), attack frequency (*p* = 0.000), hearing impairment (*p* = 0.043), aural fullness (*p* = 0.002), headache (*p* = 0.000), photophobia (*p* = 0.000), phonophobia (*p* = 0.000), PTA (*p* = 0.000), glycerin test result (*p* = 0.000), ECochG (*p* = 0.000), HST (*p* = 0.000), caloric test (*p* = 0.000), o-VEMP (*p* = 0.000), horizontal gain (*p* = 0.000), and horizontal phase (*p* = 0.000) were significant indicators in distinguishing between VM and MD (Table 2).

### 3.2. Variables That Could Differentiate VM and MD

Multivariable double logistic regression analysis of the clinical and auxiliary examination parameters demonstrated that attack frequency, phonophobia, ECochG, HST, o-VEMP, and VAT (horizontal gain) could help differentiate between VM and MD (Table 3). In order to evaluate the sensitivity and specificity of selected parameters in the differentiation between VM and MD, sensitivity, specificity, accuracy, positive PV, and negative PV were tabulated for various parameters (Table 4).

### 3.3. Predictive Variable Models for Differentiation of VM from MD

On the basis of statistically significant parameters screened by the Chi-square test and further multivariable double logistic regression analysis, we established regression equations. Regression equation P = 1/[1 + e^−(−2.269^
^× X5 − 2.395^
^× X13 + 2.141^
^× X18 + 3.949^
^× X23 + 2.798^
^× X27 − 4.275^
^× X30(1) − 5.811^
^× X30(2) + 0.873)^] (where P is predictive value and e is natural logarithm) was established to predict the differential diagnosis between VM and MD. As shown in the flow, if *p* ≥ 0.314, the predictable diagnosis is MD; if *p* < 0.314, the predictable diagnosis is VM. (Figure 1). The diagnosis point or threshold (predictive boundary value) 0.314 was obtained by the ROC curve. The area under the curve (AUC) of ROC curves of attack frequency, phonophobia, HST, ECochG, o-VEMP, and horizontal gai, were all greater than 0.5 (Figure 2).

The following method was used for the regression equation: (1) collect patients’ clinical and auditory-vestibular function features, including attack frequency, phonophobia, HST, ECochG, o-VEMP, and horizontal gain of VAT. (2) If attack frequency is less than 3 times in one month, X5 = 0. If attack frequency is greater than 3 times in one month, X5 = 1. If patients have no phonophobia, X13 = 0. If patients have phonophobia, X13 = 1. If the results of HST, ECochG, and o-VEMP are normal, the value of X18, X23, and X27 is 0; if abnormal, the value is 1. If the horizontal gain of VAT is normal, the value of X30(1) is 1, and X30(2) is 0. If the horizontal gain of VAT is subnormal, the value of X30(1) is 0, and X30(2) is 0. If the horizontal gain of VAT is paranormal, the value of X30(1) is 0, and X30(2) is 1. Then, calculate the *p* value. (3) As shown in the flow, if *p* ≥ 0.314, the predictable diagnosis is MD; if *p* < 0.314, the predictable diagnosis is VM.

The area under the curve (AUC) of ROC curves of attack frequency, phonophobia, HST, ECochG, o-VEMP, and horizontal gain were all greater than 0.5.

### 3.4. Nomograph and Applet, as Two Visualization Tools

On the basis of the results of the logistic regression, we determined the predictive factors for diagnosis between VM and MD. To facilitate the differentiation between VM and MD, we screened out six characteristic variables and then constructed a nomographic risk prediction model (Figure 3). In addition, we inputted the value of each single risk factor of the patients through the formula module of Excel software (Appendix A) and then calculated the specific value of diagnosis by means of a regression equation. The model tools made it easy for clinicians to differentiate VM and MD.

The following method was used for the nomogram: (1) the patient-related values were input on the diagnostic variable axes. (2) An intersection line was drawn perpendicular to the score axis from the marked points of each variable axis to obtain the corresponding scores of each variable. (3) Finally, all scores were added, the corresponding points on the total score axis were determined, and a vertical line was drawn perpendicular to the probability axis from the six points on the total score axis. The value obtained from the intersection is the probability of differential diagnosis between VM and MD.

### 3.5. Internal Validation of the Prediction Model

The development and validation of the predictive model was according to the requirements of the TRIPOD (Transparent Reporting of a multi-variable prediction model for Individual Prognosis or Diagnosis) Statement [17], an international guideline specifically designed for diagnosis or prognosis. The prediction model needs the verification of samples in different periods, regions, and centers to truly reflect the real prediction efficiency [16]. Therefore, we prospectively collected the clinical data of new patients as the validation population of the center in different periods, verified the model, and evaluated its clinical application value. A total of 28 patients with VM and 28 patients with MD were prospectively recruited as the validation set. Moreover, there was no significant difference in clinical features between the model and the validation group (*p* > 0.05). The internal validation was performed by evaluating the performance of the model with respect to its discriminative and calibrating ability.

### 3.6. Discriminative Power

The area under the receiver operating characteristic curve (AUROC) is commonly used in clinical practice to quantify the predictive model discrimination power. In the two groups, ROC curves were plotted. In model group, AUROC was 0.982 (95% CI: 0.968~0.996), and the cutoff value was 31.4% (Figure 4). In the validation group, AUROC was 0.880 (95% CI: 0.789~0.970).

### 3.7. Calibrating Ability

The calibration ability refers to the consistency between the predicted probability and the actual probability. The 80% CI and 95% CI of the GiViTI calibration belt did not cover the 45° diagonal bisector line, suggesting the model has good discriminative power. The Hosmer-Lemeshow test was conducted for the prediction model in the model group, with *x*^2^ = 4.562 and *p* = 0.727. In the validation group, *x*^2^ = 5.534 and *p* = 0.372. The Hosmer-Lemeshow test yielded a *p* > 0.05 of the prediction model in the two groups, indicating that the difference was not statistically significant (Figure 5).

The 80% CI and 95% CI of the GiViTI calibration belt did not cover the 45° diagonal bisector line. The Hosmer-Lemeshow test was conducted for the prediction model in the model group, with *x*^2^ = 4.562 and *p* = 0.727. In the validation group, *x*^2^ = 5.534 and *p* = 0.372.

## 4. Discussion

In this study, by examining demographic data, clinical history, and auxiliary examination results in a large cohort containing 110 MD patients and 110 VM patients, we identified features that could help distinguish between MD and VM. First and foremost, we found a profile of differentiating (predictive) factors, including attack frequency (X5), phonophobia (X13), ECochG (X18), HST (X23), o-VEMP(X27), and VAT (horizontal gain) (X30). Secondly, we, for the first time, proposed a mathematical model (a regression equation) to help differentiate between VM and MD, i.e., P = 1/[1 + e^−(−2.269^
^× X5 − 2.395^
^× X13 + 2.141^
^× X18 + 3.949^
^× X23 + 2.798^
^× X27 − 4.275^
^× X30(1) − 5.811^
^× X30(2) + 0.873)^] (where P is the predictive value and e is the natural logarithm). Finally, we established two visualization tools, i.e., a nomograph and an applet, to facilitate the clinical application of the model. In addition, the internal validation was conducted by evaluating the performance of the model with respect to its discriminative and calibrating abilities, and the results suggested that the diagnostic algorithm could achieve better diagnostic efficiency and could improve the differential diagnosis between VM and MD in clinical practice.

### 4.1. Clinical Symptoms

Previous studies showed that older age at illness onset and male sex favored MD, whereas younger age at illness onset and female sex favored VM [1,5,7,18], which was similar in our study. Some major symptoms of MD, such as fluctuating hearing loss, tinnitus, and aural fullness, may be found in VM patients. In addition, we found that the attack frequency in VM was about 4–8 times in one month, and MD about 2–3 times in one month. Nonetheless, in VM, hearing loss is unlikely to progress to a more severe condition [19]. Similarly, migraine headache, photophobia, and even migraine auras are common during Meniere attacks [20]. Although 51% of MD patients have migraine [21], 52.5% of headaches reportedly occurred concomitantly with vestibular symptoms, a rate similar to the rate found in our study (46.7%) [22]. Moreover, headaches of VM patients tend to be more severe and intermittent and last for about 11.16 days per month [22]. Hearing loss is more frequent in MD than that in VM: 81.8% and 46.7%, respectively. Similarly, Lopez-Escamez et al. found that 77.3% of MD patients, 26.3% of VM patients, and 15.4% *p*VM patients had hearing loss during attacks [18]. Comparison of accompanying symptoms between MD and VM revealed that tinnitus, aural fullness, and hearing loss were less common in patients with VM than in their MD counterparts. The overlapping pathologies of VM and MD might be ascribed to the inflammation of the trigeminal nerve vessel system, which impairs inner ear function. The VM damage predominantly affects the vestibular region, causing neuritis and vasoconstriction expansion [23]. According to its proposed mechanism, symptoms of VM could be linked to a generally “hyperexcitable” brain [24]. However, the MD principally involves a “fragile” inner ear [25].

Photo- and phonophobia were more frequent in VM patients than in MD patients. Our study found that 77.3% of VM patients and 13.6% of MD patients had phonophobia, playing an important role in the model. The sensitivity, specificity, and accuracy of the phonophobia as predictive factors in the model were 86.4%, 77.3%, and 81.8%, respectively. Beh SC et al. reported that phonophobia was as high as 90.1% in VM [26], close to the rate in our study. Ghavami et al. showed that 51% of MD patients had phonophobia [21]. Phonophobia might be an independent symptom of MD, which was found in 84% of MD patients and was unrelated to the presence of migraine [27]; this finding is inconsistent with the results of our study. Phonophobia might be attributed to lowered hearing threshold, which results from higher brainstem auditory neurol sensitivity [28]. This mechanism may explain our finding that phonophobia was infrequent in MD patients.

In addition, in our study, we found anxiety and depression were slightly more common among patients with VM than MD; however, there was no statistically significant difference between two groups. The patients’ treatment is often unsatisfactory because of the close interactions between vestibular, psychiatric, and neurological disorders. Therefore, clinicians need to consider psychological issues when treating VM or MD patients and realize their different presentations to provide appropriate treatment according to the nature of the disease.

### 4.2. Auditory Function Results

Of our MD patients, 70.9% and 80.9% yielded positive results with the glycerin test and EchochG, respectively. The higher positive rate might be ascribed to the fact that our hospital is a large general hospital, and most of the patients had serious MD. In general, the results of this test are more likely to be negative at the very early and very late stages of MD [29]. A systematic review and meta-analysis of the role of EChochG in MD diagnosis indicated that the diagnostic specificity was 83.8% [30]. The specificity of EChochG in our cohort was 87.3%, which was comparable to the aforementioned results. The glycerin test and ECochG are two traditional tests that indirectly detect the endolymphatic hydrops (EH). Although EH is a common pathology shared by MD and VM, as exhibited by gadolinium-contrasted MRI, MD patients had a substantially higher rate of EH than VM patients [11,31]. The glycerin test and EChochG are believed to be the most convenient, widely used, and specific tests for the diagnosis MD. Although contrast-enhanced MRI can visualize EH with high sensibility and specificity, some of its limitations restrict its extensive application, since it is costly, time-consuming, and not readily available in all hospitals.

### 4.3. Vestibular Function Findings

The head-shaking test (HST) is induced by oscillating the head at high frequency in the horizontal plane. It is used for both peripheral and central vestibular disorders. It is important to diagnose vestibular dysfunction; we routinely perform the horizontal HST [32]. The abnormal rate in MD was about 39.8%, and that in VM was about 14.5%, much lower than previously reported: 71% in MD, and 50% in VM [12]. This may be attributed to patients usually visiting their doctor in interval episodes. In addition, the positivity rate of HSN can decrease with the development of vestibular compensation. VAT horizonal gain decreased in 86.4% of MD patients and increased in 87.3% of VM patients. VAT horizonal gain was the most important factor or contributor in our predictive model, the sensitivity and specificity of which were 92.7% and 87.3%, respectively. VAT is a testing technique that examines the head and eye movements at high frequencies (2 to 6 Hz). VAT can provide supplementary information to other tests for VOR assessment. The pathways responsible for the VOR are very complex and principally consist of vestibular ganglia, vestibular nuclei, and oculomotor nuclei. The peripheral damage renders the primary reflex pathway incomplete, thereby down-regulating VOR and reducing the gain. If the central vestibule is structurally abnormal, the inhibitory action of the vestibular nuclei will be weakened, leading to VOR hyperfunction and gain increase. In addition, Mert Cemal Gökgöz [33] confirmed that horizontal phase values were sensitive markers for discriminating decompensated MD from compensated MD. The vertical high-frequency VOR plays an important role in visual stabilization during daily activities, such as ambulation. A high vertical phase value, in the range of 4 to 5 Hz, was associated with presence of migraine.

We found that the o-VEMP was abnormal in 25.5% of VM patients and 74.7% of MD patients. Similarly, Taylor et al. [34] and Salviz et al. [35] reported a significantly higher AR (asymmetry ratio) of ACS (air-conducted sound) cVEMP amplitudes in patients with unilateral MD (46% and 29%) as compared to those with VM (16% in both studies). VEMPs are short-latency manifestations of vestibuloocular and vestibulocollic reflexes that originate from the utricle and saccule. Thus, VEMPs have mostly been applied to the diagnosis of disorders involving the peripheral vestibular system. The high AR may be attributed to the distended saccular membrane in contact with the stapes footplate or probably to a progressive loss of vestibular hair cells and primary vestibular neurons. Moreover, Baier and Dieterich [36] found no difference in ACS cVEMPs between VM and MD. Histopathological analyses in MD have shown that, compared to the utricle, the saccule is affected more often by ELH. Furthermore, the trigeminovascular system branches into the labyrinthine arteries of both labyrinths, and the distribution might result in symmetrical involvement of both labyrinth organs in VM.

In our cohort, the caloric test yielded abnormal results in 54.5% of MD patients and 30.9% of VM patients; pathological vHIT occurred in 63.6% of MD patients and 58.7% of VM patients. Both vHIT and the caloric test can evaluate the VOR of the horizontal semicircular canal. The semicircular canal paralysis was detected by the caloric test more frequently and was more severe in MD patients than their VM counterparts; no significant difference in vHIT was found between them [37]. Blödow and colleagues compared the results of the caloric test and vHIT in 53 patients with VM and MD and found that vHIT was more sensitive for the diagnosis of peripheral hypofunction. Blödow considered that the reason for the inconsistent results was that the caloric test detects the low frequency of the semicircular canal, while vHIT detects the high frequency of the semicircular canal. The vestibular victims of MD may be frequency dependent [38]. In addition, McGarvie reported that due to endolymphatic hydrops, MD patients’ membranous semicircular canal expands. When stimulated by the caloric test, the endolymph forms local circulation in the expanded membranous semicircular canal, which reduces the flow of endolymph on both sides of crista ampullaris and reduces the deviation of crista ampullaris. During vHIT, the enlarged membranous semicircular canal did not affect the internal lymph flow caused by head flick, so the result was normal [39].

Differentiation between MD and VM faces various challenges, even if relevant diagnostic criteria are available. MD is now diagnosed on the basis of the consensus proposed by the Bárány Society [1]. In terms of VM, the Bárány Society and the International Headache Society have established a consensus document, with diagnostic criteria, which was included in the appendix of the new International Classification of Headache Disorders (ICHD)-3 beta version, which means that VM can be deemed as an emerging entity needing further research [2]. The constantly revised diagnostic guideline and multidisciplinary international expert workshops [40] have contributed greatly to the assessment of MD and VM and their treatment.

Our study has some limitations. First, our study was conducted in a single tertiary care general hospital and caution should be exercised when extrapolating the model to other hospitals with different resources and patient sources. A multicenter and prospective study with additional clinicians can enhance the robustness of the model. Another limitation is that this study failed to cover more factors, such as the temporal relation between headache and vestibular symptoms and biological markers [41], among others. In addition, another issue left unaddressed was the endotyping of the MD subjects [42,43,44,45]. The patients were not sorted in terms of interictal phase and acute stage of VM and MD. Statistically, we performed internal validation on the lesser dataset, which partially addressed the validation issue; however, this is not as robust as validation of the predictive models using external or cross validation. These are problems we will address in our future studies.

## 5. Conclusions

In this study, we for the first time worked out a predictive model for distinguishing between MD and VM. Attack frequency, phonophobia, EChochG, HST, o-VEMP, and VAT (horizontal gain) may form a battery to differentiate the two conditions. The predictive model had high sensitivity and specificity, good discriminative power, and good calibrating ability in both model and validation groups. The use of nomographs in combination with applets also facilitated the doctor–patient communication, and the intuitive display improves the diagnostic efficiency. Our study paves the way for future commodity studies of VM and MD.

## Figures and Tables

**Figure 1 jcm-11-04745-f001:**
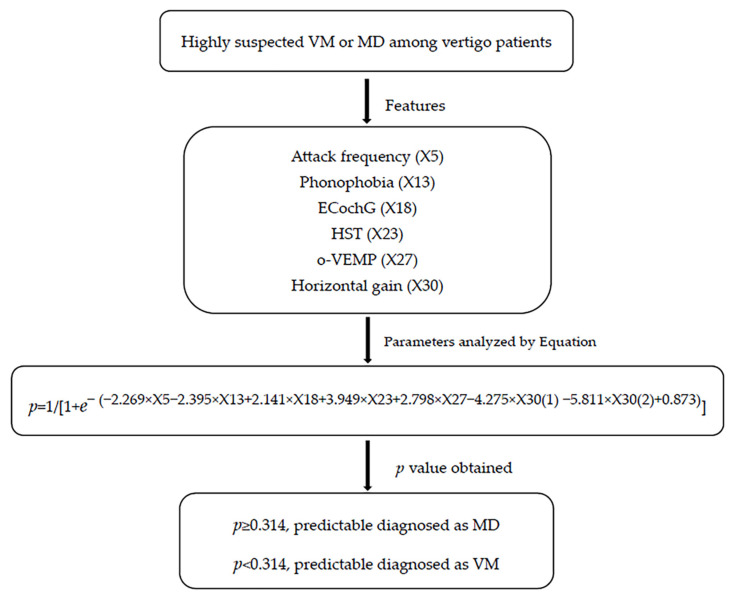
Differential diagnostic flow between VM and MD based on clinical and auditory-vestibular function features. EcochG, electrocochleogram; HST, head shake test; o-VEMP, Ocular Vestibular Evoked Myogenic Potential; VM, vestibular migraine; MD, Meniere’s disease.

**Figure 2 jcm-11-04745-f002:**
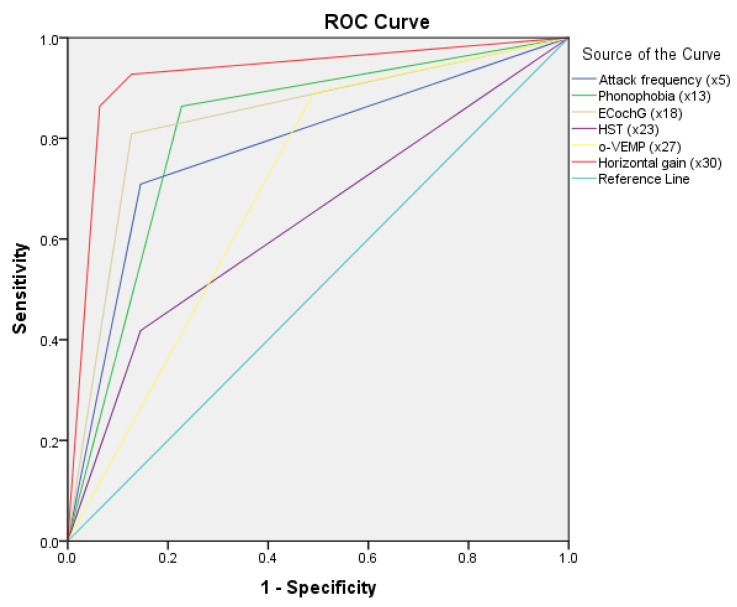
Receiver operating characteristic (ROC) curves of logistic model and specificity variables. EcochG, electrocochleogram; HST, head shake test; o-VEMP, Ocular Vestibular Evoked Myogenic Potential; VM, vestibular migraine; MD, Meniere’s disease.

**Figure 3 jcm-11-04745-f003:**
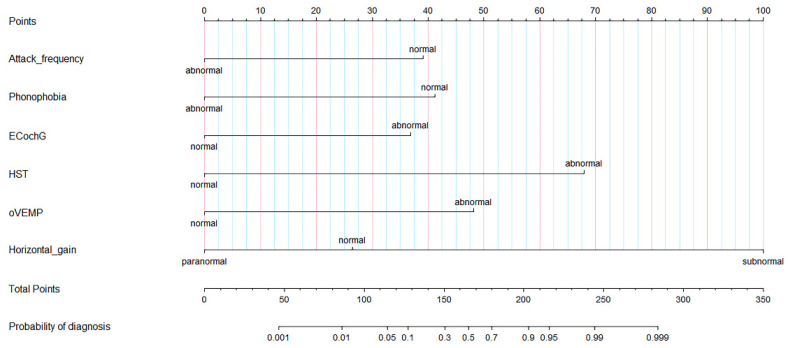
Diagnostic nomogram estimated by clinical and auditory-vestibular function features for patients with VM and MD. EcochG, electrocochleogram; HST, head shake test; o-VEMP, Ocular Vestibular Evoked Myogenic Potential; VM, vestibular migraine; MD, Meniere’s disease.

**Figure 4 jcm-11-04745-f004:**
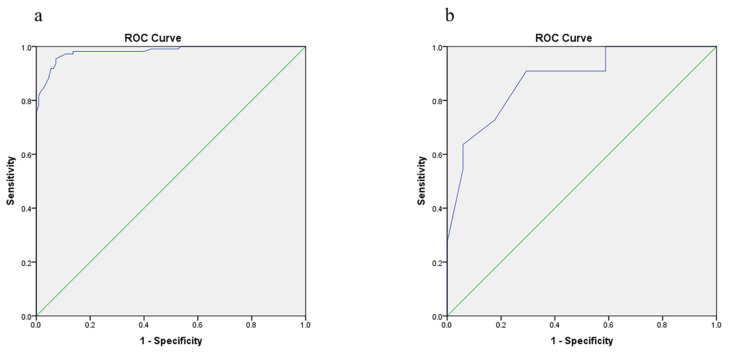
AUROC curve used to estimate the discriminative power the prediction model. (**a**) Model construction group; (**b**) validation group. AUROC, Area Under Receiver Operating Characteristic.

**Figure 5 jcm-11-04745-f005:**
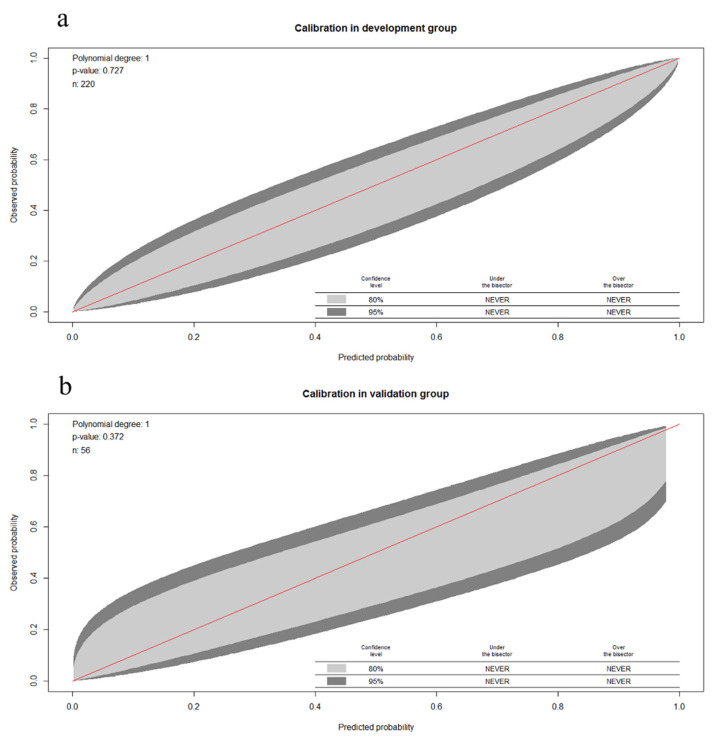
GiViTI calibration belt used to estimate the calibrating ability of the prediction model. (**a**) Model construction group; (**b**) validation group. GiViTI, Gruppo Italiano per la Valutazione degli Interventi in Terapia Intensiva.

**Table 1 jcm-11-04745-t001:** Methodology and parameters of the tests.

Test	Equipment	Method	Parameter	Valuation
PTA	Madsen Electronics Orbiter 922 Version 2 Clinical Audiometer (Otometrics A/S, Taastrup, Denmark)	Patients wear earphones attached to the audiometer. Pure tones of a specific frequency and volume are delivered to one ear at a time. The patient is asked to signal when hearing a sound. Average hearing thresholds were expressed at 125–4000 and 8000 Hz.	Thresholds and frequencies	0 = normal 1 = abnormal It was taken as abnormal if: PTA > 20 dBHL
OAE	Capella MADSEN company (Otometrics A/S, Taastrup, Den-mark)	A small probe is placed in the ear canal. This probe delivers a low-volume sound stimulus into the ear. The cochlea responds by producing an otoacoustic emission that travels back through the middle ear to the ear canal.	Otoacoustic emission is or not evoked.	0 = normal 1 = abnormal It was taken as abnormal if: not evoked
Stapedius reflex	OTOFLE100 (Otometrics A/S, Taastrup, Denmark)	Dynamic changes result from contraction of stapedius in response to stimuli of 500, 1000, 2000, and 4000 Hz at intensities of 70–115 dB sound pressure level.	Thresholds for activation.	0 = normal 1 = abnormal It was taken as abnormal if: not evoked
Glycerin test	Madsen Electronics Orbiter 922 Version 2 Clinical Audiometer (Otometrics A/S, Taastrup, Denmark)	PTA test is performed before the administration of glycerol and then patient is administered a solution of 86% of glycerol (1.5 mg/kg of body weight) dissolved in equal volume of physiological saline. PTA is then repeated at 1, 2, and 3 h of glycerol administration.	PTA threshold shift and speech discrimination	0 = normal 1 = abnormal It was taken as abnormal if: (1) hearing threshold is lowered at least 15 dB at minimum three frequencies; (2) a total pure tone threshold shift of 25 dB at three consecutive frequencies; (3) 16% improvement in speech discrimination
ECochG	Nicolet Compass Meridian (nicolet compass, U.S.A)	A sticker electrode is placed on the forehead, and foil-covered earphones are inserted into the ear canals. An audio stimulus is presented to the patient through the earphones. An electrode picks up cochlear activity that occurs in response to the sound.	Summating potential/action potential (SP/AP) amplitude ratio	0 = normal 1 = abnormal It was taken as abnormal if: an SP/AP amplitude ratio greater than 40–45%
Spontaneous nystagmus/Gaze test/Saccadic pursuit/Optokinetic test	VisalEyes^TM^ VNG, Micromedical Technologies Inc., Chatham, IL, USA	Spontaneous nystagmus: The patient looks straight ahead without focusing and is observed for nystagmus. Gaze test: the patient follows aim, so she/he is looking 30° to the right, left, up, and down. There is a pause of 20 s in each of these positions to observe for nystagmus. Saccadic pursuit: the patient follows a slowly moving aim horizontally and then vertically (from center to 30° right and then to 30° left). Optokinetic test: an individual tracks (pursuit movement) a moving object with their eyes.	Induced different type of nystagmus	0 = normal 1 = abnormal It was taken as abnormal if: patient’s eyes are observed for nystagmus
HST	/	The patient’s eyes are observed for nystagmus immediately after a passive rapid head shaking along a vertical axis at about a frequency of 2 Hz, for 20 cycles.	Induced different type of nystagmus	0 = normal 1 = abnormal It was taken as abnormal if: patient’s eyes are observed for nystagmus
Dix-Hallpike/Roll test	VisualEyes^TM^ VNG, Micromedical Technologies Inc., USA	Patient sits on a couch. Examiner holds the patients head, turns it 45° to the right, and then places the patient in a supine position so that the head hangs 30° below the horizontal. The test is repeated with head turned to left and then again in straight head-hanging position. It often used to check for a common type of vertigo called BPPV.	Induced different type of nystagmus	0 = negative 1 = positive It was taken as positive if: patient’s eyes are observed for nystagmus
Caloric test	Air caloric irrigator system (Air Fx from Micromedical Technologies Inc., USA)	VNG is used to record eye movements during the caloric test. Before the test, the ear, especially the eardrum, is checked. A small amount of cold/warm air is gently delivered into ears. The temperature of the warm and cool air is 50 °C and 24 °C, respectively.	Unilateral weakness (UW)	0 = normal 1 = abnormal It was taken as abnormal if: |UW%| greater than 25%
c-VEMP	Eclipse system (from Interacoustics A/S, Middelfart, Denmark)	Participants are asked to sit on a chair and rotate heir head to the contralateral side to activate SCM muscles. An active electrode is placed on the upper third of the ipsilateral SCM muscles, a ground electrode is put on the forehead, and a reference electrode is put on the sternoclavicular junction. Stimuli are produced by a customized VEMP software package (OtoAccess, from Interacoustics A/S, Middelfart, Denmark).	The amplitude asymmetry ratio (AR) and peak-to-peak cVEMP amplitude	0 = normal 1 = abnormal It was taken as abnormal if: (1) The AR is greater than 36%; (2) the peak-to-peak cVEMP amplitude is absent or reduced
o-VEMP	Eclipse system (from Interacoustics A/S, Middelfart, Denmark)	Participants sit on a chair and are instructed to stare up at a red spot fixed on the wall at midline in front of them. The stare forces the participants to elevate and maintain their gaze up to approximately 30° above the horizontal plane during each session of the test. An active electrode is positioned on the contralateral inferior oblique muscles, a ground electrode is applied on the forehead, and a reference electrode is placed on the chin.	Biphasic wave-form and amplitude asymmetry ratio (AR)	0 = normal 1 = abnormal It was taken as abnormal if: (1) Biphasic waveform was absent after at least 50 responses; (2) the AR is greater than 40%
v-HIT	ICS Impulse system (GN Otometrics, Denmark)	Subject wears a pair of tightly-fitting goggles equipped with video oculography camera to record and analyze the eye movement. Patient is seated upright facing the wall 1.0 m away and is instructed to fixate on a static target on the wall. The patient’s head is passively and randomly rotated to the left and right with a low amplitude (5~15°) and at a high peak velocity (150~250°/s) in an abrupt, brief and unpredictable manner. At least 20 head impulses are delivered in each direction.	Horizontal vHIT gain and re-fixation saccades	0 = normal 1 = abnormal It was taken as abnormal if the horizontal vHIT gain is <0.8 and saccades appear
VAT	Software package (VATPLUS^®^) from WSR (Western System Research, Pasadena, CA, USA)	The patient is required to fix the eyes on a target 120 cm away and asked to perform head rotations on horizontal and vertical planes. Velocity is set at 0.5–0.9 Hz in the first 6 s, and it gradually rises from 1 to 6 Hz in the next 12 s.	The gain, phase, and asymmetry are recorded at the frequencies of 2.0–6.0 Hz	0 = subnormal 1 = normal 2 = paranormal It was taken as abnormal if: Gain: The ratio of eye to head speed is <1: subnormal; the ratio of eye to head speed >1: paranormal; the ratio of eye to head speed close to 1: normal. Phase: The response time delay. Symmetry of left and right eye velocity (normal <±10%)
SOT	SMARTEquitest platform (NeuroCom International Inc., Clackamas, OR, USA)	Participants stand on a SMART Equitest platform and are asked to stand upright and maintain balance during the test. There are six sensory conditions (SOT1-SOT6).	Vestibular (VEST ratio) = SOT5/SOT1	0 = normal 1 = abnormal It was taken as abnormal if: VEST ratio < 0.577; it was taken as normal if: VEST ratio > 0.577

Abbreviations: PTA, pure tone audiometry; OAE, otoacoustic emissions; ECochG, electrocochleogram; VNG, videonystagmography, HST, head-shaking test; BPPV, benign paroxysmal positional vertigo; VEMPs, vestibular evoked myogenic potentials; v-HIT, video-head impulse test; VAT, vestibular auto-rotation test; SOT, sensory organization test.

**Table 2 jcm-11-04745-t002:** Differentiation of vestibular migraine (VM) and Meniere’s disease (MD).

Parameter	VM (*n* = 110)	MD (*n* = 110)	*p*-Value	Code	Valuation
Demographic features					
Gender(male/female) ratio	44/66	65/45	0.120	X1	0 = female, 1 = male
Age (range) year	50.18 ± 13.318	48.95 ± 12.457	0.310	X2	continuous variable
Clinical features					
Vertigo/dizzy (%)	98.2	95.4	0.840	X3	0 = none 1 = yes
Illness duration (%)	7.1/17.9/32.1/42.9	0/9/45.5/45.5	0.002 *	X4	0 =< 7 d, 1 = 7 d~30 d, 2 = 1 m~1 y, 3 => 1 y
Attack frequency (%)	14.9/85.1	71.5/29.5	0.000 *	X5	0 =< 3 times 1 => 3 times
Visual motion (%)	61.8	59.1	0.754	X6	0 = none 1 = yes
Nausea and vomiting (%)	63.6	72.7	0.281	X7	0 = none 1 = yes
Hearing impairment (%)	46.7	81.8	0.043 *	X8	0 = none 1 = yes
Tinnitus (%)	57.3	63.6	0.334	X9	0 = none 1 = yes
Aural fullness (%)	10.9	23.7	0.002 *	X10	0 = none 1 = yes
Headache with vestibular episodes (%)	47.3	13.6	0.000 *	X11	0 = none 1 = yes
Photophobia (%)	73.0	14.2	0.000 *	X12	0 = none 1 = yes
Phonophobia (%)	77.3	13.6	0.000 *	X13	0 = none 1 = yes
Auditory-vestibular function					
PTA (%)	45.5	80.9	0.000 *	X14	0 = normal 1 = abnormal
OAE (%)	80	73.6	0.263	X15	0 = normal 1 = abnormal
Stapedius reflex (%)	1.8	9.1	0.100	X16	0 = negative 1 = positive
Glycerin test (%)	14.5	70.9	0.000 *	X17	0 = negative 1 = positive
ECochG (%)	12.7	80.9	0.000 *	X18	0 = normal 1 = abnormal
Spontaneous nystagmus (%)	12.7	18.2	0.382	X19	0 = negative 1 = positive
Gaze test (%)	3.6	1.0	0.450	X20	0 = normal 1 = abnormal
Saccadic pursuit (%)	3.6	2.4	0.579	X21	0 = normal 1 = abnormal
Optokinetic test (%)	9.7	13.6	0.070	X22	0 = normal 1 = abnormal
HST (%)	14.5	39.8	0.000 *	X23	0 = normal 1 = abnormal
Dix-Hallpike (%)	32.7	13.6	0.054	X24	0 = negative 1 = positive
Roll test (%)	21.8	9.1	0.064	X25	0 = negative 1 = positive
Caloric test (%)	30.9	54.5	0.000 *	X26	0 = normal 1 = abnormal
o-VEMP (%)	25.5	74.5	0.000 *	X27	0 = normal 1 = abnormal
c-VEMP (%)	23.6	31.8	0.296	X28	0 = normal 1 = abnormal
v-HIT (%)	58.7	63.6	0.170	X29	0 = normal 1 = abnormal
VAT (Horizontal gain) (%)	6.4/6.4/87.3	86.4/6.4/7.3	0.000 *	X30	0 = subnormal, 1 = normal, 2 = paranormal
VAT (Horizontal phase) (%)	3.6/27.3/69.1	1.3/58.1/40.6	0.000 *	X31	0 = subnormal, 1 = normal, 2 = paranormal
VAT (Vertical gain) (%)	0/94.5/5.5	1/93/6	0.184	X32	0 = subnormal, 1 = normal, 2 = paranormal
VAT (Vertical phase) (%)	0/94.5/5.5	6/92/2	0.184	X33	0 = subnormal, 1 = normal, 2 = paranormal
VAT (Asymmetry) (%)	12.7	13.6	0.879	X34	0 = normal 1 = abnormal
SOT (Vestibular) (%)	60.0/40.0	77.3/22.7	0.140	X35	0 = normal 1 = abnormal
Radiologic					
MRI (%)	10.9	13.6	0.634	X36	0 = normal 1 = abnormal
Rating Scale					
PHQ9 (%)	32.7	18.2	0.071	X37	0 = normal 1 = abnormal
GAD7 (%)	21.8	9.1	0.064	X38	0 = normal 1 = abnormal
SCL90 (%)	14.5	13.6	0.884	X39	0 = normal 1 = abnormal
SSS (%)	80.0	90.0	0.630	X40	0 = normal 1 = abnormal

Note: * *p* < 0.05. Abbreviations: MD, Meniere’s Disease; VM, vestibular migraine; PTA, pure tone audiometry; OAE, optoacoustic emission; ECochG, electrocochleogram; VNG, videonystagmography; HST, head-shaking test; VEMPs, vestibular evoked myogenic potentials; v-HIT, video-head impulse test; VAT, vestibular autorotation test; SOT, sensory organization test; SSS, somatic self-rating scale; GAD-7, generalized anxiety disorder; PHQ-9, Patient Health Questionnaire-9.

**Table 3 jcm-11-04745-t003:** Multivariate logistic regression analysis of clinical and auditory-vestibular function features in patients with VM and MD.

Indexes (Variable Code)	B	SE	Wald *X*^2^	*p*-Value	OR (95% CI)
Attack frequency (X5)	−2.269	0.979	5.376	0.020	0.103 (0.015–0.704)
Phonophobia (X13)	−2.395	0.900	7.076	0.008	0.091 (0.016~0.532)
ECochG (X18)	2.141	0.859	6.206	0.013	8.505 (1.578~45.828)
HST (X23)	3.949	1.317	8.986	0.03	51.861 (3.923–68.531)
o-VEMP (X27)	2.798	0.901	9.643	0.002	16.405 (2.806~95.898)
Horizontal gain (X30)	−4.458	1.008	19.569	0.000	0.012 (0.002~0.084)
Constant	0.873	1.252	0.486	0.486	2.394

Abbreviations: ECochG, electrocochleogram; HST, head-shaking test; o-VEMP, ocular vestibular evoked myogenic potential; B, regression coefficient; SE, standard error; *X*^2^, chi-square value; OR, odds ratio; CI, confidence interval.

**Table 4 jcm-11-04745-t004:** Sensitivity, specificity, accuracy, positive PV, and negative PV tabulated for significant parameters.

Features	Sensitivity	Specificity	Accuracy	Positive PV	Negative PV
Attack frequency	70.9	85.5	78.2	83	74.6
Phonophobia	86.4	77.3	81.8	79.2	85
ECochG	80.9	87.3	84.1	86.4	82.1
HST	58.2	85.5	71.8	80	67.1
o-VEMP	89.1	50.9	70.0	64.5	82.4
Horizontal gain	92.7	87.3	90	87.9	92.3
Diagnostic model	93.3	94.5	95.9	94.7	93.2

Abbreviations: ECochG, Electrocochleogram; HST, Head-shaking test; o-VEMP, Ocular vestibular evoked myogenic potential; PV, predicted value.

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
