# Peer review of "Development and Validation of the Predictive Model for the Differentiation between Vestibular Migraine and Meniere’s Disease"

_jcm, 2022, doi:10.3390/jcm11164745_

Round 1

Reviewer 1 Report

Some typos:

Line 121: "Head impulse test (HST)", it should be "(HIT)"

Line 141: "OAE, Optoacoustic emission", it should be Otoacoustic emissions

Line 142 HST, "Head-shaking nystagmus", it should be better "Head-shaking test" 

Line 173: HST , here it appears that "HST" is correct.

Line 191: HST, Head-shaking nystagmus, it would be better "Head-shaking test" 

Line 196: HST, Head-shaking nystagmus 

Line 351: Head-shaking nystagmus (HSN) 

Use HST for "Head-shaking test" or HSN if you prefer  "Head-shaking nystagmus". Although the terms are synonymous, to clarify, use the same term along the text.

Reviewer 2 Report

Interesting and valuable work. Appreciate the authors acknowledge patients may have both VM and MD. Seems like VAT and vHIT results should have been more in agreement. Unsure how many clinicians still using VAT. 

There are some minor English language issues throughout that should be addressed.

Reviewer 3 Report

In this paper, the Authors investigate clinical and instrumental features between Vestibular Migraine and Meniere disease, searching for a simple and affordable method to differentiate the two pathologies. 

The research is well conducted, and the number of tests performed s very impressive for a retrospective study performed in a single tertiary care general hospital. The statistical analysis is very significant even if the results don't show substantial differences in the knowledge between both pathologies.

The value of this work is in the have demonstrated statistically that the exams are useful for differential diagnosis.

Some clarification and corrections are needed:

1) Line 144 Change Vestibular autoro-tation test in Vestibular auto-rotation test

2) TABLE 1: Why have you used for the caloric test 50°C and 24°C grade of temperature instead 30°C and 44°C?   

3)LINE 110   It is not clear if the validation group was taken from the study group or not; clarification is needed.

4) The results of the evaluation scales were not presented, and it would be better to eliminate them from the materials and methods or add the results and discuss them.

Author Response

This manuscript is a resubmission of an earlier submission. The following is a list of the peer review reports and author responses from that submission.

Round 1

Reviewer 1 Report

I would like to congratulate the authors for developing this predictive model to differentiate vestibular migraine from Meniere’s disease.

To contribute, I would suggest some corrections:

1 – It is very important to describe the phase of the diseases when all assessments were done. We must know it If we want to replicate this research. Moreover, the results can be different if we evaluate patients during the crises or out of crises, or if it is done in early or advanced phases of the same disease.

2 – the data from Table 1 are misaligned and difficult to correlate among columns

3 – At lines 288 and 289 the word phonophobia appears repeatedly.

Reviewer 2 Report

This is a mixed retrospective-prospective study assessing ways to differentiate between Meniere’s Disease (MD) and vestibular migraine (VM). The authors of the study retrospectively analysed clinical data on MD and VM patients, validated them prospectively and then proposed a tool helpful in differential diagnosis. The aim of this study is highly relevant for clinical practice since currently no reliable ways to discern between MD and VM are in use.

Merits

The group of patients included in the study should provide enough statistical power for most interventions in homogenous groups.

The proposed mechanism (applet) for differential diagnosis seems to be an attractive option for future use in clinical practice.

Critique

MAJOR

1.       Patients selection is not explained adequately:

a.       Considering that the study aims at differentiating between similar disorders, what were the overlapping VM and MD symptoms that lead to exclusion and why?

b.       Authors classify patients as heaving headache or migraine but it is unclear what this terms mean in this setting (who made the diagnosis and on what basis?).

c.       According to authors collected data did not include photophobia, phonophobia and aura symptoms. However, in results photophobia and phonophobia occur. Aura is not mentioned throughout the manuscript although it is one of the key diagnostic conditions of VM.

d.       Missing data seems to play an important role in this study. Up to 2/3 of parameters might have not been collected. However there is no information in results on what were the numbers used for statistical analysis.

2.       It is surprising that authors do not undertake discussion with current guidelines on differentiating between VM and MD (i.e. 10.3389/fneur.2021.812678). Especially, considering that their results differ with some studies on VEMP mentioned in these guidelines.

3.       Parts of the manuscript are difficult to comprehend due to language used (perseverations, inadequate phrasing).

4.       There is no statement regarding patients consent to participate in prospective part of the study.

MINOR

1.       Abstract needs a thorough reediting to explain clearly study design and results. Currently it needlessly concentrates on statistical methods. Also references in abstract should be avoided.

2.       It should be noted that International Headache society does not recognise VM as a standalone diagnosis (it is included in appendix as a ‘candidate’ awaiting ‘confirmation’).

3.       Table 1. is misaligned – it is unclear which value belongs in which line.

4.       Parts of the first chapter in ‘Results’ belongs to ‘Material and Methods’.

5.       Authors sometimes use ‘migraine’ as a synonym for ‘headache’ (e.g. p1 l17)

6.       Abbreviations should be explained when first used.

Reviewer 3 Report

There are some major questions that should be explained before considering the publication of the manuscript.

Major concerns:

-      -  We do not know if bilateral MD patients are included in the study.

-       - Family history is not reported.

-       - Autoimmune disease history is not reported.

-       - Only headache is specified as during a vestibular episode. We do not know if other clinical variables (hearing loss, tinnitus, phonophobia and so on) were considered during an attack or in the interictal period. 

-       The authors have explored spontaneous nystagmus, gaze, OKN and saccadic pursuit, but not head impulse test, head-shaking nystagmus nor vibration-induced nystagmus; surprisingly, these later tests are mentioned in the introduction. This is relevant as head-shaking nystagmus and vibration-induced nystagmus are the more common vestibular signs in patients with MD. Moreover,some studies have noted that in patients with VM 50 % had abnormal head-shaking nystagmus and 32 % abnormal vibration-induced nystagmus (Wang, 2012).

-       The major concerns are related to the methods used to study the patients as the authors do not detail the methods used for the audio-vestibular function tests:

o    How is hearing impairment defined? Do you distinguish between unilateral and bilateral hearing loss or fluctuating/permanent? 

o   What testing parameters do you consider in ECoG: latencies and amplitudes of SP and AP, SP/AP amplitude ratio? 

o   What parameters do you use to consider VEMPs as abnormal: threshold, amplitude, latency, interaural amplitude difference? 

o   What value of canal paresis is the cut-off in your laboratory? 

o   What canals do you explore with vHIT: horizontal only, vertical also? You consider a vHIT abnormal based in gain, saccades, or both?

o   Laboratory tests (cytokine profile, autoimmune, hormones) are not considered; in this sense, Flook et al. 2019 paper is not cited for discussion.

o   Only the existence of cerebral infarction is considered on MRI.

o   The table 1 is very important but it is mpossible to analyze as the lines do not correspond with the parameters of the left column. 

Minor concerns:

-       Line 36: peripheral vestibular disorder is much better than peripheral otologic disorder.

-       Lines 84 and 87: in the original Bárány consensus paper the word “jointly” was used because the criteria were “jointly” formulated by the Classification Committee of the Bárány Society and some other scientific societies.

-       Lines 213-216: the verb appears missing in this sentence.

-       Lines 235-236 duplicate lines 229-230.

-       Lines: 288-289: “phonophobia and phonophobia”. 

-       Lines 287-298: phonophobia and hyperacusis are different symptoms. Patients with MD suffer from phonophobia, hyperacusis or both?

-       Lines 331-332: “AR” and “ACS” have not been defined previously along the text.

Round 2

Reviewer 1 Report

The authors have answered all raised questions and I am satisfied with their answers. 

Author Response

Dear reviewer,

        We thank you for your careful and encouraging evaluation of our manuscript (ID: jcm-1778240). There is no doubt that these comments from you are very valuable and helpful for revising and improving our manuscript. We would like to thank you once again for your work in the processing of our paper.

Sincerely yours,

Su-Lin Zhang, M.D., Ph.D.,

Department of Otorhinolaryngology, Union Hospital, Tongji Medical College, Huazhong University of Science and Technology, No. 1277 Jiefang Avenue, Wuhan City, Hubei Province, P. R. China, 430022.

Reviewer 2 Report

The authors sufficiently addressed issues raised in my review.

Author Response

(The authors gave the same response as above.)

Reviewer 3 Report

Unfortunately, the answer to the questions on the methodology (equipment, parameters) of the tests carried out is unsatisfactory. The authors limit themselves to giving a generic description of the different tests.

We still don’t know what thresholds and frequencies they use to define hearing loss, they do not explain when the glycerol test is positive, the value to consider low gain in vHIT in each canal is not specified nor what type of equipment they use for vHIT or VEMPs. We do not know how they perform the sensory organization test and when they consider the result as abnormal.
